# Abuse recognition by shelter staff and shelter animal adopters

**Ineke R. van Herwijnen**◉°*, **Nadieh Reinders**◉°, **Esmee M. Bus**°, **Claudia M. Vinke**◉‡

Division of Animals in Science and Society, Department of Population Health Sciences, Faculty of Veterinary Medicine, Utrecht University, Utrecht, The Netherlands

◉ These authors contributed equally to this work.
‡ CMV also contributed equally to this work.
* i.r.vanherwijnen@uu.nl

## Abstract

Science has provided insight into the physical signs of animal abuse, such as the presence of fractures in different healing stages. Possible behavioural signs of animal abuse are understudied. In studies on child abuse, behavioural signs have been identified. We aimed to study if similar signs would be viewed as relevant signs of animal abuse, focussing on physical abuse of cats and dogs. We targeted shelter staff and shelter animal adopters, as these people may come in contact with abused animals, are known to be willing to participate in research and may do so without ethical constraints applying. We found that the behavioural signs based on child abuse studies, were deemed relevant for the assessment of animal abuse by the respondents in our small-scale study (N = 23 shelter staff and N = 132 shelter animal adopters). Behavioural signs of fear, such as fear of the owner, were deemed most relevant. Person-specific fear was indicated by shelter staff and shelter animal adopters as a more accurate indicator of animal abuse than aggression, problems with human-social bond formation and person-specific pleasing behaviour. This study held a limited number of respondents, but indicates a possible use of behavioural signs in studying animal abuse. To further assess the validity of animal behavioural signs as a source of animal abuse recognition, future studies will need to observe animals known to have previously suffered abuse. These studies should aim to use ethograms that include objective descriptions of behaviours that may be of interest to animal abuse experiences. Our study indicates that shelter animal staff and adopters may differ in their appraisal of behaviours other than fear, when assessing a likeliness of animal abuse. Studying such differences in larger samples may provide insight into how these populations assess an animal's possible abuse experience or lack thereof.

**Data availability statement:** All relevant data are within the paper and its Supporting information files.

**Funding:** The author(s) received no specific funding for this work.

**Competing interests:** The authors have declared that no competing interests exist.

## Introduction

Animals under direct human care may suffer, if the needs or preferences of their human caretaker are different from their needs. This is seen for instance in humans preferring a physical appearance in animals that comes with discomfort and ailments [1–5]. Suffering may also be caused by animal abuse [6–13]. Animal abuse is prevalent in our societies [14–16], as seen in the reporting thereof by 6% of 55,000 French higher education students [14], with domestic cats and dogs among the animals on which veterinarians report abuse [15]. Animal abuse can take several forms, with one form regarding physical abuse. Physical animal abuse was defined as: 'active, physical, with serious violence or direct harm to animals including fractures induced by kicking, injuries induced by falling, burning, or performing home surgery' in a South Korean study [16, p.5]. Multiple definitions of physical animal abuse exist [6–13,17–20], however in our current study we use the definition of physical abuse regarding the infringement of an animal's bodily integrity, such as by hitting or kicking an animal [17–20]. To identify animal abuse, an animal's physical signs may be used [21–25]. Telling physical signs of animal abuse have been documented, indicating that for example the injury type, age, healing stage and the location of the injury offer valuable information for abuse recognition [6,7,10,13,17,20–30].

A timely and adequate recognition of animal abuse is of importance to animal welfare [31], just as it is important in child abuse cases [32,33]. In child abuse cases, timely recognition is aided by the availability of behavioural signs indicative of child abuse [33]. Examples of behavioural signs that aid early recognition of child abuse are wariness of adult contact and displaying behavioural extremes from aggressiveness to withdrawal [33]. So far, in studies on animal abuse the behavioural signs of animal abuse have received less scientific attention than the physical signs of animal abuse [11,15,34,35]. This may be due to the fact that physical signs, if still present when an animal is seen by a veterinarian, can be assessed in the veterinary clinical setting. However, assessing behavioural signs of animal abuse may be challenging in the veterinary clinical setting, as animals may for instance show fear behaviours in this setting, consequential to previous painful experiences during veterinary treatment [35,36], hampering the assessment of other fear causes, such as animal abuse. Consequently, the veterinary clinical setting is a less logical candidate setting for a study on behavioural signs of animal abuse. We know of no studies assessing or comparing settings that may allow us to study behavioural signs of animal abuse. As an alternative to the veterinary clinical setting, the shelter setting could potentially offer an opportunity to study behavioural signs of animal abuse. This, as shelters may harbour animals that were previously abused, such as when seized, which can be on different grounds, including abuse [37]. Also, after acclimatisation to a shelter surrounding, a broader range of behaviours in animals may be seen than in the veterinary clinical setting [38]. Yet, we found no studies into the recognition of behavioural signs of animal abuse in sheltered animals during shelter stay or post adoption, although we point at the possibility that such data exist, e.g., in data sources not covered in the scientific literature. Gathering knowledge on animal abuse in the shelter setting may in part depend on the shelter staff involved with the animals and

the shelter animal adopters. Hence, information on how animal abuse and behavioural signs thereof are viewed by these populations may be of value to assess the extent to which they may form a reliable study source, in addition to for example objective observations of the animal behaviour in more experimental settings.

The apparent lack of studies on behavioural signs of animal abuse, makes it relevant to study their possible use in identification thereof. We aimed to determine if behavioural signs of child abuse [33,39] are viewed as relevant by shelter staff and shelter animal adopters to assess if an animal likely suffered animal abuse. We did so by assessing in these populations: 1) if animal behavioural signs are perceived as relevant to assess possible previous animal abuse experiences and 2) how specific animal behavioural signs are viewed for a relevance to animal abuse assessment. We have two reasons for involving shelter staff and animal adopters. The first reason is that abused animals form a part of the animal population under their care [37]. This makes for an accessible population that may have under their care the animals of interest. Other such populations, are difficult to reach, due to it for example being unsafe or unethical to survey domestic abuse victims who have or had animals under their care, or to study animals seized, but not released and possibly involved in a legal trajectory. Also, deceased animals cannot be studied for behaviour, which is a contrast to studies assessing physical abuse signs. The second reason is that shelter staff and adopters may have views on possible previous abuse of animals under their care. These views are of interest as presently little is known on the validity of such views in regard to appointing an animal's previous abuse or lack thereof. Learning more about these views can help us to not wrongfully appoint or reject a history of animal abuse.

Our current study focused on domesticated cats and dogs. We hypothesised that shelter staff and shelter animal adopters would point at the behaviours of person-specific fear, such as an animal seeming frightened by the owner, more so than other-than-fear-behaviours, specifically the behaviours of a) aggression, b) problems with human-social bond formation and c) person-specific pleasing behaviour. This, as generally trauma events like animal abuse are associated with fear experiences which may form fear memory [40,41]. We hypothesised that shelter staff would report more so than shelter animal adopters that behavioural signs are a likely source for indicating animal abuse. This, as shelter staff education may come with higher knowledge levels of the cat and dog species, including their behaviour. Thus, we aim with our study to create an initial step in identifying behavioural signs of abuse that are noticeable to human observers. We stress that our research does not evaluate the accuracy of human assumptions in relation to the actual animal experiences.

## Methods

### Ethical considerations

This project was part of a larger project for which we sought and received ethical approval from the Science-Geosciences Ethics Review Board of Utrecht University for involved human respondents on October 1st, 2024 (ERB Review Science-24–0094). We approached Dutch shelter personnel via email and shelter animal adopters via social media channels, asking them to participate of their own free will, after reading information on the study and providing informed consent in the survey module by clicking a survey button that indicated 'I consent', before continuing to the survey. The topic of this study regarded an emotionally burdening topic, which we specifically addressed in the respondent information. No minors were included in the study.

### Respondent recruitment and survey

We recruited Dutch *shelter personnel* between April 25th, 2025 and June 2nd, 2025 by emailing Dutch shelters, following up via telephone within a week. The shelters were asked to distribute the invitation to partake to personnel. The list of shelters was based on shelter listings from the Dutch Association for the Protection of Animals (Nederlandse Vereniging tot Bescherming van Dieren) and the NFDO (Nederlandse Federatie Dierenhulporganisaties) and complemented by searching the internet for shelters, not listed by these two organisations. We recruited Dutch *shelter animal adopters* between April 25th, 2025 and June 2nd, 2025 via social media channels. In the message we explained the purpose of the study,

specifically requesting people not to partake if they were known to be sensitive to (animal) abuse topics, in the interest of their own wellbeing. Both samples, the shelter personnel and the shelter animal adopters, were asked to participate of their own free will, after reading information on the study. They provided informed consent by clicking on a survey button that indicated 'I consent'. Alternatively, if they did not want to partake, they closed their browser after reading the information on the survey. The topic of this study regarded an emotionally burdening topic, which we specifically addressed in the respondent information. Only adults (18 years and up) were invited to participate.

The survey's main part regarded questions on animal abuse. (See S1 File for the complete list of survey questions). The questions were derived from one source on animal abuse [35] and two sources on child abuse [33,39]. As our study regarded a first step towards behavioural signs of animal abuse only, we aimed to include as many of the behavioural signs known from child abuse to assess how our study samples would rate their likely application to animal abuse situations. Thus, our only exclusion criterium was a listed sign not regarding visible behaviour, such as 'feels deserving of punishment'. The source on animal abuse [35] held a direct question on behavioural signs of animal abuse, which we presented to our respondents. The two sources on child abuse ( [33], with the original source being: Prevent Child Abuse Georgia. Recognizing physical abuse: http://www.preventchildabusega.org/html/physabuse.html and [39]), allowed for selection of behavioural signs of abuse. As we selected only the signs that regarded visible behaviours, we dropped three items from [33] and six items from [39] as shown in S2 File. We then grouped three items from the latter study of 'acting out', 'inappropriate anger', 'aggressive behaviour' under 'aggression' and integrated overlapping items from these two studies (see S2 File). We selected the signs scored at higher accuracy (<2.5 mean accuracy score, with the mean ranging from 1.8 to 3.7) as assessed by child professionals [39]. We translated the signs to Dutch language after adapting the signs to reflect an animal sign instead of a child sign, replacing for instance the word 'child' for 'animal' as shown in S2 File. We reversed the coding for 'accuracy'. This seven-point Likert-scale ranged from very accurate ('1') to very inaccurate ('7') in the original source [39]. We deemed reversal necessary as to avoid confusion in our Dutch respondents, that are accustomed to lower numbers associating with lower values.

## Data preparation and statistical analysis

Data from Qualtrics® was gathered in Microsoft® Excel. Statistical analysis of the data was performed with SPSS® for Windows®, version 29 (SPSS Inc, Chicago, USA). The median and ranges of responses are reported. Additionally, means ± standard deviations (SD) are presented to facilitate comparison with human literature.

We tested our first hypothesis that participants would point at the behaviours of person-specific fear more so than on behaviours other than fear behaviour, with Wilcoxon signed-rank tests. Here we compared the accuracy scores for the fear behaviour 'expressing person-specific fear' to other than fear behaviours of 'aggression', 'problems with forming a social bond with humans' and 'expressing person-specific overly 'pleasing' behaviour through for instance submissive behaviour/appeasement'. We tested our second hypothesis that shelter staff would report more so than shelter animal adopters that behavioural signs are a likely source of indicating animal abuse, with Mann-Whitney U tests. We regarded P-values of <0.05 as significant.

## Results

### Respondent characteristics

To indicate the size of the shelters where our respondents (shelter staff) worked, we asked for the estimated number of animals yearly entering the shelter. Dogs came into shelters at lower numbers than cats, with most shelters (52.2%, N = 12) caring for up to 50 dogs and most shelters (39.1%, N = 9) caring for between 250–500 cats. We also asked if shelters *generally* gathered or received information on animals in their care suffering animal abuse. Sources of such information could be a previous owner, the police/inspection/community bodies, or other sources. Of the partaking shelter staff, 8.7% (N = 2) indicated never to receive such information, 21.7% (N = 5) indicated to receive such information from previous

owners, 82.6% (N = 19) (also) from the police/inspection/community bodies, 56.5% (N = 13) (also) from other sources. Our study sample of 23 shelter staff respondents consisted mainly of females (78.3%, N = 18; males: 17.4%, N = 4; other/prefer not to say: 4.4%, N = 1) and most worked as an animal caretaker (cat and/or dog: 60.8%, N = 14; and/ or as shelter manager, involved in care also: 26.1%, N = 6); see S3 Table for all details.

Our study sample of 132 shelter animal adopters consisted also of mainly females (91.7%, N = 121; males: 6.1%, N = 8; other/prefer not to say: 2.3%, N = 3). Of the respondents most (69.7%, N = 92) indicated ownership of a dog, 30.3% (N = 40) of a cat; see S3 Table for all details. We asked the shelter animal adopters if the rehoming organisation had provided information on their animal having suffered animal abuse. For 67.4% (N = 89) the suffering of animal abuse was not indicated at adoption. When asking the adopters if they thought their animal was abused before it came to live with them, 25.0% (N = 33) thought this not to be the case, 36.4% (N = 48), thought this with certainty to be the case and 36.4% (N = 48) with a likeliness. We asked the adopters to indicate if they thought their animal was mistreated through one or more forms of mistreatment, allowing them to provide multiple answers. This rendered a total of 182 answers, indicating that a part of the sample deemed their animal to have suffered multiple forms of mistreatment. Physical animal abuse (36.8%, N = 67) and mental animal abuse (36.3%, N = 66) were indicated most often. Neglect was indicated for 26.9% (N = 49) of the animals.

## Shelter staff's views on behavioural signs of animal abuse

To the question 'Based on an animal's behaviour, do you think you can identify whether it is being mistreated by its owner?', shelter animal adopters responded in accord. The majority of the respondents indicated that this is *sometimes* possible: 65.2% (N = 15). 34.8% (N = 8) indicated that this is possible *in most cases.* None indicated that this was not possible or not to know.

When assessing which behavioural signs shelter staff would regard as possibly indicative of animal abuse, 'An animal seems frightened by owner(s) or of going towards them' scored the highest median value of 5 (range 3–5) on a scale of 1 ('very unlikely') to 5 ('very likely'), as shown in Table 1. A median 'somewhat likely' score (4, ranges varying from 1/2/3–5) was for 'An animal is wary of adult contact', 'An animal seems frightened by going home', 'An animal displays a vacant or frozen gaze', 'An animal displays behavioural extremes, such as aggressiveness', 'An animal displays behavioural extremes, such as withdrawal' and 'An animal sits or lies very still while surveying surroundings'. None of the signs scored a median 1/2 ('[very/somewhat] unlikely') score.

In line with this, when assessing deemed *accuracy* of behavioural signs of animal abuse, 'Expressing person-specific fear' scored a median 7 ('very accurate'), on a scale of 1 ('very inaccurate') to 7 ('very accurate'), as seen in Table 2. Shelter staff deemed person-specific behaviours highly accurate, as seen in a median score of 6 for 'Anxiety when a particular

**Table 1. Shelter animal staff's view on the *likeliness* of a behavioural signs of animal abuse.**

| | Median | Min | Max | Mean | SD |
|---|---|---|---|---|---|
| An animal seems frightened by owner(s) or of going towards them. | 5 | 3 | 5 | 4.7 | 0.6 |
| An animal is wary of adult contact. | 4 | 1 | 5 | 4.0 | 1.0 |
| An animal seems frightened by going home. | 4 | 2 | 5 | 3.7 | 0.8 |
| An animal displays a vacant or frozen gaze. | 4 | 3 | 5 | 3.8 | 0.7 |
| An animal displays behavioural extremes, such as aggressiveness. | 4 | 1 | 5 | 3.8 | 1.0 |
| An animal displays behavioural extremes, such as withdrawal. | 4 | 3 | 5 | 4.0 | 0.7 |
| An animal sits or lies very still while surveying surroundings. | 4 | 2 | 5 | 3.8 | 1.0 |
| An animal seems less capable of forming a social bond with humans. | 3 | 1 | 5 | 3.3 | 1.1 |
| An animal acts apprehensive when a human (adult or child) cries. | 3 | 1 | 5 | 2.8 | 1.0 |
| An animal uses demanding (manipulative) behaviour to get attention. | 3 | 1 | 5 | 2.7 | 1.1 |
| A young animal shows inappropriate or precocious maturity. | 3 | 1 | 5 | 2.9 | 1.0 |

**Table 2. Shelter animal staff's view on the *accuracy* of behaviour signs of animal abuse.**

| | Median | Min | Max | Mean | SD |
|---|---|---|---|---|---|
| Expressing person-specific fear | 7 | 1 | 7 | 5.7 | 1.7 |
| Anxiety when a particular person is near | 6 | 1 | 7 | 5.4 | 1.8 |
| Reluctance to be *in the presence* of a particular person | 6 | 1 | 7 | 5.5 | 1.8 |
| Reluctance to be *left alone with* a particular person | 6 | 1 | 7 | 5.5 | 1.8 |
| Anxiety | 5 | 1 | 7 | 4.4 | 2.1 |
| Aggression | 5 | 1 | 7 | 4.1 | 2.0 |
| Hypervigilant behaviour | 5 | 1 | 7 | 4.4 | 1.9 |
| Guarded behaviour | 5 | 1 | 7 | 4.1 | 2.0 |
| Reluctance to go to a particular *place/ room* | 5 | 1 | 7 | 4.5 | 2.0 |
| Sudden behavioural change, which cannot be explained otherwise | 5 | 1 | 7 | 4.5 | 1.9 |
| Problems with forming a social bond with humans | 5 | 1 | 7 | 4.3 | 2.0 |
| Withdrawal | 4 | 1 | 7 | 3.9 | 2.0 |
| Depression | 4 | 1 | 7 | 3.9 | 2.0 |
| Auto mutilation | 4 | 1 | 7 | 3.9 | 1.8 |
| Behavioural problems | 4 | 1 | 7 | 4.1 | 2.1 |
| Insecure behaviour | 4 | 1 | 7 | 4.2 | 2.0 |
| Behavioural avoidance | 4 | 1 | 6 | 4.0 | 1.8 |
| Expressing *person-specific* overly 'pleasing' behaviour through for instance submissive behaviour/ appeasement | 4 | 1 | 7 | 4.3 | 1.9 |
| Acting overly 'pleasing' through for instance submissive behaviour/ appeasement | 3 | 1 | 7 | 3.5 | 1.9 |

person is near', 'Reluctance to be *in the presence of* a particular person' and 'Reluctance to be *left alone with* a particular person' (all ranges 1–7). None of the signs scored a median of 1 or 2.

We compared distribution between accuracy scores for the fear behaviour 'Expressing person-specific fear' to other than fear behaviours of 'Aggression', 'Problems with forming a social bond with humans' and 'Expressing *person-specific* overly 'pleasing' behaviour through for instance submissive behaviour/appeasement' with Wilcoxon signed-rank tests and found significantly higher accuracy scores for 'Expressing person-specific fear' than for each of the three other behaviours (Wilcoxon signed-rank, aggression: z=−7.2, P<0.001; social bond formation: z=−7.0, P<0.001; person-specific pleasing: z=−6.9, P<0.001).

### Shelter animal adopters' views on behavioural signs of animal abuse

To the question 'Based on an animal's behaviour, do you think you can identify whether it is being mistreated by its owner?', shelter animal adopters responded in accord. A near similar percentage of the respondents indicated that this is *sometimes* possible: 48.5% (N=64), as possible *in most cases:* 47.8% (N=64). 3.0% (N=4) indicated this was not possible, 0.8% (N=1) not to know.

When assessing which behavioural signs shelter animal adopters would regard as possibly indicative of animal abuse, alike shelter staff, 'An animal seems frightened by owner(s) or of going towards them' scored the highest median value of 5 (range 1–5) on a scale of 1 ('very unlikely') to 5 ('very likely'), as shown in Table 3. Also alike shelter staff, a median 'somewhat likely' score (4, ranges 1–5) was for the behaviours of: 'An animal is wary of adult contact', 'An animal seems frightened by going home', 'An animal displays a vacant or frozen gaze', 'An animal displays behavioural extremes, such as aggressiveness', 'An animal displays behavioural extremes, such as withdrawal' and 'An animal sits or lies very still while surveying surroundings'. Unlike shelter staff the behaviour of 'An animal seems less capable of forming a social bond with humans' in shelter adopters also scored a median 4 (range 1–5). None of the signs scored a median 1/2 ('[very/ somewhat] unlikely') score.

**Table 3. Shelter animal adopters' view on the *likeliness* of a behavioural signs of animal abuse.**

| | Median | Min | Max | Mean | SD |
|---|---|---|---|---|---|
| An animal seems frightened by owner(s) or of going towards them. | 5 | 1 | 5 | 4.4 | 0.9 |
| An animal is wary of adult contact. | 4 | 1 | 5 | 4.1 | 1.0 |
| An animal seems frightened by going home. | 4 | 1 | 5 | 3.8 | 1.0 |
| An animal seems less capable of forming a social bond with humans. | 4 | 1 | 5 | 3.7 | 1.1 |
| An animal displays a vacant or frozen gaze. | 4 | 1 | 5 | 3.8 | 1.0 |
| An animal displays behavioural extremes, such as aggressiveness. | 4 | 1 | 5 | 3.7 | 1.0 |
| An animal displays behavioural extremes, such as withdrawal. | 4 | 1 | 5 | 4.0 | 0.9 |
| An animal sits or lies very still while surveying surroundings. | 4 | 1 | 5 | 3.7 | 1.0 |
| An animal acts apprehensive when a human (adult or child) cries. | 3 | 1 | 5 | 2.8 | 1.0 |
| An animal uses demanding (manipulative) behaviour to get attention. | 3 | 1 | 5 | 2.7 | 1.1 |
| A young animal shows inappropriate or precocious maturity. | 3 | 1 | 5 | 3.0 | 0.9 |

When assessing deemed *accuracy* of behavioural signs of animal abuse, anxiety- and reluctance-related behaviours, as well as person-specific fear scored a median 6 (all ranges 2–7), on a scale of 1 ('very inaccurate') to 7 ('very accurate'), as seen in Table 4. Differently than the shelter staff, in shelter adopters a median 6 (2–7) was also scored for 'Sudden behavioural change, which cannot be explained otherwise' and the adopters scored a higher median than the shelter staff for 'Anxiety' and 'Reluctance to go to a particular place/room'. The twelve other behaviours all scored a median 5 (2–7).

We compared distribution between accuracy scores for the fear behaviour 'Expressing person-specific fear' to other than fear behaviours of 'Aggression', 'Problems with forming a social bond with humans' and 'Expressing *person-specific* overly 'pleasing' behaviour through for instance submissive behaviour/appeasement' with Wilcoxon signed-rank tests and found significantly higher accuracy scores for 'Expressing person-specific fear' than for each of the three other behaviours

**Table 4 . Shelter animal adopters' view on the *accuracy* of behaviour signs of animal abuse.**

| | Median | Min | Max | Mean | SD |
|---|---|---|---|---|---|
| Anxiety | 6 | 2 | 7 | 5.4 | 1.4 |
| Anxiety when a particular person is near | 6 | 2 | 7 | 6.0 | 0.9 |
| Reluctance to be in the *presence of* a particular person | 6 | 2 | 7 | 6.0 | 1.0 |
| Reluctance to be *left alone with* a particular person | 6 | 2 | 7 | 6.0 | 1.1 |
| Reluctance to go to a particular *place/ room* | 6 | 2 | 7 | 5.4 | 1.3 |
| Expressing person-specific fear | 6 | 2 | 7 | 6.1 | 1.0 |
| Sudden behavioural change, which cannot be explained otherwise | 6 | 2 | 7 | 5.3 | 1.3 |
| Withdrawal | 5 | 2 | 7 | 5.0 | 1.6 |
| Depression | 5 | 2 | 7 | 4.8 | 1.4 |
| Auto mutilation | 5 | 2 | 7 | 4.8 | 1.5 |
| Aggression | 5 | 2 | 7 | 4.9 | 1.4 |
| Behavioural problems | 5 | 2 | 7 | 5.1 | 1.5 |
| Hypervigilant behaviour | 5 | 2 | 7 | 4.9 | 1.5 |
| Guarded behaviour | 5 | 2 | 7 | 5.0 | 1.5 |
| Insecure behaviour | 5 | 2 | 7 | 4.7 | 1.6 |
| Behavioural avoidance | 5 | 2 | 7 | 5.2 | 1.3 |
| Acting overly 'pleasing' through for instance submissive behaviour/ appeasement | 5 | 2 | 7 | 4.7 | 1.4 |
| Expressing *person-specific* overly 'pleasing' behaviour through for instance submissive behaviour/ appeasement | 5 | 2 | 7 | 5.2 | 1.4 |
| Problems with forming a social bond with humans | 5 | 2 | 7 | 5.0 | 1.5 |

**Table 5. Comparison of shelter staff's and animal adopters' views on behavioural signs of animal abuse.**

|  | Staff (%; N) | Adopters (%; N) | Total of the two samples (%; N) |
|---|---|---|---|
| Yes, in most cases | 34.8% (8) | 47.7% (63) | 45.8% (71) |
| Yes, sometimes | 65.2% (15) | 48.5% (64) | 51.0% (79) |
| No | 0% (0) | 3.0% (4) | 2.6% (4) |
| Don't know/unsure | 0% (0) | 0.8% (1) | 0.6% (1) |
| Total | 100% (23) | 100% (132) | 100% (155) |

**Table 6. Significantly different accuracy ratings for behavioural signs of animal abuse by shelter staff and animal adopters.**

|  | Staff (median, range) | Adopters (median, range) | Mann-Whitney U test (z, P-value) |
|---|---|---|---|
| Acting overly 'pleasing' through for instance submissive behaviour/ appeasement | 3 (1-7) | 5 (2-7) | z = −3.04, P = 0.002 |
| Behavioural avoidance | 4 (1-6) | 5 (2-7) | z = −2.77, P = 0.01 |
| Withdrawal | 4 (1-7) | 5 (2-7) | z = −2.54, P = 0.01 |
| Depression | 4 (1-7) | 5 (2-7) | z = −2.19, P = 0.03 |
| Behavioural problems | 4 (1-7) | 5 (2-7) | z = −2.14, P = 0.03 |
| Auto mutilation | 4 (1-7) | 5 (2-7) | z = −2.06, P = 0.04 |

(Wilcoxon signed-rank, aggression: z = −3.5, P < 0.001; social bond formation: z = −3.3, P = 0.001; person-specific pleasing: z = −3.5, P < 0.001).

### Differences between shelter staff and animal adopters in views on behavioural signs

We studied if shelter staff would report higher possibility of identifying animal abuse through behavioural signs than shelter animal adopters, by firstly comparing their responses to the question 'Based on an animal's behaviour, do you think you can identify whether it is being mistreated by its owner?'. We found shelter staff indicating that this is possible *in most cases,* less so (34.8%) than shelter animal adopters (47.7%), and we found them indicating more so (65.2%) than shelter animal adopters (48.5%), that this is *sometimes* possible, although both reported high rates of accordance, as shown in Table 5 (Mann-Whitney U test, z = −0.99, P = 0.32). Only in the group of adopters, were respondents indicating that this is not possible or to not know/unsure, and this at a low percentage (4%; N = 5).

Secondly, we assessed possible differences between shelter staff and shelter animal adopters in deemed accuracy of nineteen signs and found with Mann-Witney U tests significant differences for six behaviours, as indicated in Table 6. Notably, shelter staff rated accuracy generally lower than shelter animal adopters.

## Discussion

We studied shelter staff's and shelter animal adopters' views on behavioural signs of animal abuse, through a survey based on behavioural signs of child abuse. Assessing how behavioural signs of child abuse may translate to animal abuse is of relevance as to date behavioural signs of animal abuse form an understudied scientific topic [11]. In this first step towards future studies with an ultimate aim to establish valid behavioural signs of animal abuse, we found promising results, which merit further studying, -deploying methods of stronger scientific strength-, of how behavioural signs may contribute to the recognition of animal abuse.

Our hypothesis that shelter staff and shelter animal adopters would point at the behaviours of person-specific fear more so than the other-than-fear-behaviours, was supported by our findings. Both shelter staff and shelter animal adopters

indicated a higher accuracy for person-specific fear behaviour than for the behaviours of a) aggression, b) problems with human-social bond formation and c) person-specific pleasing behaviour. Person-specific fear was scored at the highest accuracy of 7 by shelter staff and of 6 by adopters. We opted to assess fear behaviour in contrast to these three behaviours. Fear behaviour may be viewed to a higher degree as a sign of animal abuse as generally trauma events such as abuse may form fear memory [40,41]. However, aggression was owner-reported in the form of aggression towards unfamiliar people at the second highest level in previously abused dogs, after fear of unfamiliar humans [41]. Attachment/ attention-seeking behaviours were also mentioned to a high degree in these abused dogs [41], behaviours which may be seen as contrasting to fear, if fear comes with distance increasing behaviour. Due to this contrast, participants could possibly view the latter behaviours a less likely or less accurate sign of animal abuse. The third sign, person-specific pleasing behaviour, included the 'person-specific' aspect and allowed us to compare person-specific aspects in fear and non-fear behaviour. Our substantiation of person-specific fear behaviour being rated a more accurate behavioural sign than the other three behaviours, points to the necessity of further studying if and how this behaviour, in addition to other possible behavioural signs can serve as a valid indicator of an animal suffering animal abuse. A question to address, is if alternative explanations of person-specific fear can be differentiated when assessing behavioural signs of fear, as to avoid unsubstantiated assumptions of animal abuse causality. Another question to address, is if other behavioural signs, or a combination of such signs, might get too little attention where as these may be valid indicators of abuse, as previously identified in studies on child abuse [33,39].

Our hypothesis that shelter staff report more so than shelter animal adopters that behavioural signs are a likely source of indicating animal abuse, was not substantiated. Shelter staff and shelter animal adopters partaking in this study viewed animal behaviour as a possible source of identifying animal abuse and more so than the Swedish veterinarian profession [35]. Yet, contrary to our expectations, staff were more reserved in their responses than adopters, with a lower percentage of staff than adopters, indicating that the recognition is possible in *most cases*, than it being *sometimes possible*. Our assumptions were based on a higher likelihood that shelter staff would have received education, including education on behavioural signs relating to various animal welfare aspects. Thus, we expected shelter staff to report behavioural signs a likely source of indicating animal abuse more so than shelter animal adopters. Background of shelter staff varies, however behavioural knowledge is mentioned as a critical component of shelter staff capacities [42–44]. However, we found no studies indicating if and how behavioural knowledge is addressed in shelter staff education and how possible signs of animal abuse may or may not be a part of curricula. If behavioural aspects, including the aforementioned signs, are addressed in curricula, a possible Dunning-Kruger effect may explain why shelter staff answered more reservedly than shelter animal adopters. The Dunning-Kruger effect regards an overestimation of competence in areas where one lacks competence [45,46]. This psychological effect may affect uneducated animal adopters more so than staff. Alternatively, or additionally to this possible psychological effect, shelter staff and animal adopters may have different expectations of animal abuse affecting the animals under their care, or different knowledge of an animal having suffered animal abuse. Shelter staff in our study indicated that generally information may be available on animal abuse, when animals are given in their care. They may therefore be more informed of an animal suffering animal abuse. Such information may affect their appraisal of behavioural signs. Additionally, expectations of why animals are shelter-surrendered may affect shelter staff and animal adopters differently. We know of no studies comparing such expectations between these two groups, but some studies into shelter animal adopters' experiences are available. These studies indicate prior concerns of those contemplating to adopt a cat or dog, such as regarding an unknown history of the animal or their unwanted behaviours [47]. Such concerns may be valid, as 72% of N = 192 adopted dogs reportedly showed undesired behaviour in the first week after adoption, such as aggression (24%) and fearfulness (21%) [37]. Undesired behaviour was reported at higher rates for dogs that suffered 'abuse' (no precise definition for abuse was provided in this study; [37]). After half a year the rates of reported behaviour were lower for fear, but not for aggression [37], and aggression was listed as a return reason for adopted cats and dogs [48]. Often

provided reasons to opt for adoption when contemplating to acquire a cat or dog are for it to be ethically or morally correct [49]. So, when a choice for a shelter adoption is made 'to do good' and the animal shows undesired behaviours, expectations of a rescued animal 'being grateful' and 'good' may be unfulfilled. Consequently, it may be helpful to explain such unfulfilled expectations through a previous owner mistreating the animal and behavioural signs may more readily be viewed as a sign of animal abuse. Finally, we stress that our study sample of shelter staff was small. Thus, further studying is necessary to substantiate our findings. We point at two seemingly lacking scientific insights. The first regards how shelter staff education addresses animal behavioural recognition, including recognition of possible signs of animal abuse. The second regards how shelter animal adopter view abused animals and expect their behaviours to be after adoption [50].

Our study's limitations are not only in the limited number of participants in each sample. Due to this limited number of participants, we caution against extrapolation of our findings to the total (Dutch) population of shelter staff and animal adopters. Also, we deemed it necessary to stay close to the approach taken in the child abuse studies that formed the basis of our study. This allows for the child abuse study outcomes to be compared to the outcomes of this study. Consequently, we opted not to work with objective descriptions of animal behavioural signs. Thus, there is a need for such objectively defined behavioural descriptions, specified in an ethogram, to form the basis of an objective assessment of animal abuse signs in future studies. To date, when animal abuse literature describes behavioural signs, these are often described in general terminology, without behavioural specifications, such as 'signs of distress', 'the animal displays an unexplained change in behaviour', or 'observations of normal/ abnormal behaviour' [6]. Establishing an ethogram that includes behaviours that objectively and discriminatively describe those behaviours that may be of interest as behavioural signs of animal abuse is one of the many steps to be taken in future studies. The value of such future studies will be in aiding professionals, that presently indicate that physical signs (alone) are not always sufficient as indicators of animal abuse [35], through gathering scientific information on behaviours and behavioural patterns that may differentiate between animals that did and did not suffer animal abuse and consequentially aid in (early) recognition of animal abuse, facilitate screening protocols advised for early assessment of situation of domestic violence [26,51]. Finally, shelter staff amongst other animal welfare workers have been recognised to run high risk of negative effects of their work environment, such as of moral injury and burn-out due to witnessing animal suffering [52]. Helping shelter staff in animal abuse recognition and facilitating appropriate measures, may be one route to limiting the burden that comes with these professions. As such, adding to the physical signs of animal abuse, the option of assessing behavioural signs of animal abuse, may be a driver for both animal welfare and human well-being.

Our study represents an initial step in identifying behavioural signs of abuse that are noticeable to human observers. However, it is important to clarify that our research does not evaluate the accuracy of human assumptions in relation to the actual experiences of the animal. Also, our study findings cannot be used to assess animal abuse based on behavioural signs. We stress that the surveying of our particular study populations of shelter animal staff and adopters, comes with a limited use of our results. When constituting an ethogram of behaviours that may serve as possible abuse signs, our present study should not be used to determine the inclusion or exclusion of possible behaviours. We did not work with trained observers and did not study the animals themselves. Value of our study is in having gathered a first insight into our sample's views on possible abuse signs. The findings indicate the relevance of a future possibility for such assessment. By establishing deemed relevance, we have not yet established deemed value of behavioural signs in animal abuse assessment. Behaviours of interest need to be identified and studied in observational studies of animals, preferably comparing the behaviour of animals that have suffered animal abuse to that of animals that have not suffered animal abuse. Once behavioural differences have objectively been established the behaviours will next need to be validated for their potential use to indicate animal abuse. It is likely that in these studies seized sheltered animals will be studied. Their caretakers may be asked to report on animal behaviour, including on possible behavioural signs of animal abuse. Our study highlights that shelter animal staff and adopters may differ in their recognition of behaviours other than

fear, when assessing likeliness of animal abuse. Consequently, either training caretakers involved in animal behaviour studies in their reporting of animal abuse, and/ or using additional validation sources, may benefit the quality of studies on animal abuse.

## Supporting information

**S1 File. Shelter staff and adopter questions.**
(DOCX)

**S2 File. Behavioural signs from child-directed studies on child abuse recognition.**
(DOCX)

**S3 Table. Respondent characteristics.**
(DOCX)

## Author contributions

**Conceptualization:** Ineke R. van Herwijnen, Nadieh Reinders, Esmee M. Bus.

**Data curation:** Ineke R. van Herwijnen.

**Formal analysis:** Ineke R. van Herwijnen.

**Investigation:** Ineke R. van Herwijnen, Esmee M. Bus.

**Methodology:** Ineke R. van Herwijnen.

**Project administration:** Nadieh Reinders.

**Supervision:** Ineke R. van Herwijnen.

**Writing – original draft:** Ineke R. van Herwijnen.

**Writing – review & editing:** Ineke R. van Herwijnen, Nadieh Reinders, Claudia M. Vinke.

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
