## [Decision Letter · Decision Letter 0]

22 Oct 2025

Dear Dr. Herwijnen,

We look forward to receiving your revised manuscript.

Kind regards,

Brittany N. Florkiewicz, Ph.D.

Academic Editor

PLOS ONE

Journal Requirements:

Additional Editor Comments:

I strongly encourage you to thoroughly check the grammar and spelling of your manuscript before resubmission. Several potential reviewers declined the invitation to review because they found the aims of your study unclear based on the wording of the abstract. Both reviewers also mentioned issues related to clarity and grammar.

Thank you for submitting your manuscript. I have consulted two reviewers with expertise closely aligned with the scope of your work. Both reviewers have suggested major revisions, citing issues with the clarity of the manuscript, the operationalization of key definitions, and the transparency regarding methodology and limitations.

Reviewer's Responses to Questions

**Comments to the Author**

1. Is the manuscript technically sound, and do the data support the conclusions?

Reviewer #1: Yes

Reviewer #2: Partly

2. Has the statistical analysis been performed appropriately and rigorously?

Reviewer #1: I Don't Know

Reviewer #2: I Don't Know

3. Have the authors made all data underlying the findings in their manuscript fully available?

Reviewer #1: Yes

Reviewer #2: Yes

4. Is the manuscript presented in an intelligible fashion and written in standard English?

Reviewer #1: Yes

Reviewer #2: No

Reviewer #1: This study provides original insights into a topic that has remained significantly underexamined in recent years. Its findings are expected to make a meaningful contribution to the academic discourse surrounding the detection of animal abuse. Nonetheless, revisions should be implemented to enhance the clarity of terminology and to more precisely delineate the study’s limitations

Introduction

It is unclear why the authors assert that intentional trauma infliction is not always clearly distinguished from the more passive animal neglect

For this academic paper is fundamental the definition of emotional abuse but the authors only offer an example, please define it.

It is mentioned that the veterinary clinical setting may come with challenges for assessing animal behavioral signs, highlighting shelters as an alternative. However, it does not elaborate on why shelters represent a preferable option.

Methods

Although signs of emotional abuse in children may help guide the identification of behavioral indicators in animals, it is essential to clarify the criteria used to determine which aspects of children's behavior are suitably applicable to animals.

Discussion

The difficulty in detecting behavioral indicators of animal abuse is acknowledged; therefore, it is important to clearly outline the specific challenges involved in identifying abuse based on behavioral signs.

Finally, given that the study’s results are based on the perceptions of non-experts, the limitations should clearly state that the findings do not determine the primary behavioral indicators for suspecting animal abuse.

Reviewer #2: 1) The terms used within this research are often vague or undefined. For instance, "aggression"is a multifaceted set of distinctly identifiable behaviors, many of which have additional significance based on context. This needs to be defined. "Seems wary of adult contact", "Seems less capable of forming human bonds", etc. are amorphous judgement calls not defined to the survey respondents. These characteristics are presumably also occurring in a shelter environment, which is stressful and has been clearly identified as unlikely to produce reliable, reproducable results. "Precocious maturity" in a domestic dog is an inappropriate measure - puppies do not show such behavior. Other factors in Supporting #2 are inappropriately projected from human children to dogs or have potentially ery different emotional or behavioral bases. Dogs are not humans. 2) Sample size is quite small as identified by the authors. 3) The claims of actual abuse are poorly documented. Abuse is variously interpreted by adopters and shelter staff without clear definition. Were the police referrals prosecuted? Successfully? Were the police cases assumed to be valid without prosecution or conviction? 4) Sweeping conclusions are made that may not be supported by confirmed research: i.e. "Person-specific fear behavior was regarded a more accurate indicator of animal abuse than aggression, problems with human-social bond formation and person-specific pleasing behavior." Abstract line 43-46. According to whom? 'Aggression', 'person-specific fear', and 'problems with human-social bond formation' are undefined and unsupported. 5) The expression of factors suffers from the presumably Dutch to English translation. Wording is awkward and convoluted, causing potential or real misunderstanding. I.e. "Environmental signs may regard an animal caretaker's higher alcohol consumption or a weaker belief in justice." Introduction, line 86. What? What signs? How much alcohol? Compared to whom? "This, as animals may show different suppressed, and/or fear behaviors due to fear of veterinary surroundings and previous painful experiences in the veterinary clinical setting." I understand the idea, but the wording is convoluted and may not be clear to others. Also see line 247 "...being frightened of the owner or going towards them...". If you can't go toward or away, then what can the animal do?

**Do you want your identity to be public for this peer review?** For information about this choice, including consent withdrawal, please see our Privacy Policy

Reviewer #1: No

Reviewer #2: **Yes:** James W. Crosby M.S., PhD.

---

## [Author Response · Author response to Decision Letter 1]

5 Nov 2025

Reviewer's Responses to Questions:

Reviewer #1: This study provides original insights into a topic that has remained significantly underexamined in recent years. Its findings are expected to make a meaningful contribution to the academic discourse surrounding the detection of animal abuse. Nonetheless, revisions should be implemented to enhance the clarity of terminology and to more precisely delineate the study’s limitations.

Thank you for your kind words and suggestions for improving the manuscript.

Introduction: It is unclear why the authors assert that intentional trauma infliction is not always clearly distinguished from the more passive animal neglect. For this academic paper is fundamental the definition of emotional abuse but the authors only offer an example, please define it.

Thank you for pointing out this opportunity for improving our manuscript. We have rewritten the introduction, based on other feedback. We have provided more focus by providing details on physical animal abuse only. In line with this focus, we provide a definition in lines 59 to 61: ‘‘active, physical, with serious violence or direct harm to animals including fractures induced by kicking, injuries induced by falling, burning, or performing home surgery’. Please note that we also indicate that multiple definitions exist.

It is mentioned that the veterinary clinical setting may come with challenges for assessing animal behavioral signs, highlighting shelters as an alternative. However, it does not elaborate on why shelters represent a preferable option.

We have added this elaboration in lines 80 to 82, as following; ‘Shelters may harbour animals that were previously abused, such as when seized, which can be on different grounds, including abuse [37]. After acclimatisation to a shelter surrounding, a broader range of behaviours in animals may be seen than in the veterinary clinical setting [38]’.

Methods: Although signs of emotional abuse in children may help guide the identification of behavioral indicators in animals, it is essential to clarify the criteria used to determine which aspects of children's behavior are suitably applicable to animals.

We now describe the selection approach using the specific word ‘criterium’ in lines 129 to 132, with as the main text: ‘As our study regarded a first step towards behavioural signs of animal abuse only, we aimed to include as many of the behavioural signs known from child abuse to assess how our study samples would rate their likely application to animal abuse situations. Thus, our only exclusion criterium was a listed sign not regarding visible behaviour, such as ‘feels deserving of punishment’.

Based on the reviewer’s feedback we also, additionally, stress the following. We do not regard the output of this study suitable behavioural signs for studying animal abuse. Assessing such suitability will require further studying, using different study methods than applied in this first step. Lines 379 to 391 in the discussion section: ‘Our study is a first step towards identifying behavioural signs of animal abuse. Our study findings cannot be used to assess animal abuse based on behavioural signs. The findings are a first step towards a future possibility for such assessment. By establishing deemed relevance, we have not yet established deemed value of behavioural signs in animal abuse assessment. Behaviours of interest need to be studied further in observational studies of animals, preferably comparing the behaviour of animals that have suffered animal abuse to that of animals that have not suffered animal abuse. Once behavioural differences have objectively been established the behaviours will next need to be validated for their potential use to indicate animal abuse. It is likely that in these studies seized sheltered animals will be studied. Their caretakers may be asked to report on animal behaviour, including on possible behavioural signs of animal abuse. Our study highlights that shelter animal staff and adopters may differ in their recognition of behaviours other than fear, when assessing likeliness of animal abuse. Consequently, either training caretakers involved in animal behaviour studies in their reporting of animal abuse, and/ or using additional validation sources, may benefit the quality of studies on animal abuse’

Discussion: The difficulty in detecting behavioral indicators of animal abuse is acknowledged; therefore, it is important to clearly outline the specific challenges involved in identifying abuse based on behavioral signs.

Finally, given that the study’s results are based on the perceptions of non-experts, the limitations should clearly state that the findings do not determine the primary behavioral indicators for suspecting animal abuse.

Thank you for addressing that these points could have been made more specifically in our manuscript. The text indicated in lines 379 to 391 and specified above, makes this more specific in the new version of our manuscript.

Reviewer #2: 1) The terms used within this research are often vague or undefined. For instance, "aggression" is a multifaceted set of distinctly identifiable behaviors, many of which have additional significance based on context. This needs to be defined. "Seems wary of adult contact", "Seems less capable of forming human bonds", etc. are amorphous judgement calls not defined to the survey respondents. These characteristics are presumably also occurring in a shelter environment, which is stressful and has been clearly identified as unlikely to produce reliable, reproducable results. "Precocious maturity" in a domestic dog is an inappropriate measure - puppies do not show such behavior. Other factors in Supporting #2 are inappropriately projected from human children to dogs or have potentially ery different emotional or behavioral bases. Dogs are not humans.

Thank you for your feedback and we could not agree more! This is also why we addressed in our discussion section the need for establishing an ethogram. Based on your feedback we have at several points in our manuscript highlighted more how this study is merely a first step in studying the indication of behavioural signs of animal abuse. Also, we have added in lines 379 to 391 a section which may make the limitations of our study clearer: ‘Our study is a first step towards identifying behavioural signs of animal abuse. Our study findings cannot be used to assess animal abuse based on behavioural signs. The findings are a first step towards a future possibility for such assessment. By establishing deemed relevance, we have not yet established deemed value of behavioural signs in animal abuse assessment. Behaviours of interest need to be studied further in observational studies of animals, preferably comparing the behaviour of animals that have suffered animal abuse to that of animals that have not suffered animal abuse. Once behavioural differences have objectively been established the behaviours will next need to be validated for their potential use to indicate animal abuse. It is likely that in these studies seized sheltered animals will be studied. Their caretakers may be asked to report on animal behaviour, including on possible behavioural signs of animal abuse. Our study highlights that shelter animal staff and adopters may differ in their recognition of behaviours other than fear, when assessing likeliness of animal abuse. Consequently, either training caretakers involved in animal behaviour studies in their reporting of animal abuse, and/ or using additional validation sources, may benefit the quality of studies on animal abuse.’

In addition, we have now not only indicated in the methods section that our study followed the methodology of two scientific studies on child abuse – using their terminology as a starting point. In addition to this section and the supplement with further details, we have now also added these lines 358 to 367 in the discussion section: ‘(…). Also, we deemed it necessary for this first step to stay close to the approach taken in the child abuse literature that formed the basis of this study. Thus, we did not work with detailed descriptions of behavioural signs. Although these signs could form a basis for future studies, we point at the need to further specify behaviours in an ethogram. To date, when animal abuse literature describes behavioural signs, these are often described in general terminology, without behavioural specifications, such as ‘signs of distress’, ‘the animal displays an unexplained change in behaviour’, or ‘observations of normal/abnormal behaviour’ [6]. Establishing an ethogram that includes behaviours that objectively and discriminatively describe those behaviours that may be of interest as behavioural signs of animal abuse is one of the many steps for future studies.’

2) Sample size is quite small as identified by the authors.

We have further highlighted this in the discussion section in line 358: ‘Our study’s limitations are not only in the limited number of participants in each sample.’ Note that despite substantial rewriting of the abstract, we have kept the sample sizes as to make the small-scale of the study clear to our readers at first glance.

3) The claims of actual abuse are poorly documented. Abuse is variously interpreted by adopters and shelter staff without clear definition. Were the police referrals prosecuted? Successfully? Were the police cases assumed to be valid without prosecution or conviction?

We have reworded this text in lines 167 to 168: ‘We also asked if shelters generally gathered or received information on animals in their care suffering animal abuse.’

4) Sweeping conclusions are made that may not be supported by confirmed research: i.e. "Person-specific fear behavior was regarded a more accurate indicator of animal abuse than aggression, problems with human-social bond formation and person-specific pleasing behavior." Abstract line 43-46. According to whom? 'Aggression', 'person-specific fear', and 'problems with human-social bond formation' are undefined and unsupported.

Thank you for indicating how we could improve our abstract. We have now reworded sections to adhere to your feedback. E.g. in lines 35 to 36: ‘Therefore, we aimed to study if behavioural signs from studies on child abuse are viewed as relevant by shelter staff and shelter animal adopters, (…)’ and in lines 40 to 41: ‘Person-specific fear was indicated by shelter staff and shelter animal adopters as a more accurate indicator of (…)’.

5) The expression of factors suffers from the presumably Dutch to English translation. Wording is awkward and convoluted, causing potential or real misunderstanding. I.e. "Environmental signs may regard an animal caretaker's higher alcohol consumption or a weaker belief in justice." Introduction, line 86. What? What signs? How much alcohol? Compared to whom? "This, as animals may show different suppressed, and/or fear behaviors due to fear of veterinary surroundings and previous painful experiences in the veterinary clinical setting." I understand the idea, but the wording is convoluted and may not be clear to others. Also see line 247 "...being frightened of the owner or going towards them...". If you can't go toward or away, then what can the animal do?

Thank you for highlighting the sections with convoluted sentences. The manuscript was thoroughly checked and adapted throughout.

---

## [Decision Letter · Decision Letter 1]

26 Dec 2025

PLOS One

Dear Dr. Herwijnen,

Thank you for submitting your manuscript to PLOS ONE. After careful consideration, we feel that it has merit but does not fully meet PLOS ONE’s publication criteria as it currently stands. Therefore, we invite you to submit a revised version of the manuscript that addresses the points raised during the review process.

Thank you for your patience, and I apologize for the delayed decision on your manuscript. Reviewer #2 was unable to look at your revisions (due to time constraints), so we had to secure an additional third reviewer.

Reviewer #1 is satisfied with your revisions, and has no additional feedback. Reviewer #3, however, has raised some excellent points regarding study limitations and variable descriptions. They recommend rejection, but I believe that their comments may be addressed with major revisions (i.e., adding and clarifying information).

Once you have resubmitted your manuscript, I will reach out and see if Reviewer #3 is willing to give it another look. If not, I will look over your changes myself and provide additional feedback as needed.

We look forward to receiving your revised manuscript.

Kind regards,

Brittany N. Florkiewicz, Ph.D.

Academic Editor

PLOS One

Journal Requirements:

Reviewers' comments:

Reviewer's Responses to Questions

**Comments to the Author**

Reviewer #1: All comments have been addressed

Reviewer #3: (No Response)

2. Is the manuscript technically sound, and do the data support the conclusions?

Reviewer #1: Yes

Reviewer #3: No

3. Has the statistical analysis been performed appropriately and rigorously?

Reviewer #1: I Don't Know

Reviewer #3: No

4. Have the authors made all data underlying the findings in their manuscript fully available?

Reviewer #1: Yes

Reviewer #3: Yes

5. Is the manuscript presented in an intelligible fashion and written in standard English?

Reviewer #1: Yes

Reviewer #3: Yes

Reviewer #1: Research on emotional abuse in animals is essential for detecting and prosecuting cases of animal cruelty. The authors have strengthened the manuscript and successfully addressed all of my comments.

Reviewer #3: The authors set out to examine an important and inadequately studied question in animal sheltering and veterinary medicine - How do we best accurately recognize behavioral signs of physical abuse in companion dogs and cats in a shelter setting, where it is likely abused animals may be found? They authors acknowledge that their small study is a first step towards developing a scientifically valid ethogram of the behavior of physically abused dogs and cats. I have significant concerns about the methods by which the authors sought to answer this question, however, which forms the basis of opinion that this manuscript should be rejected.

1. The participants in the study, shelter workers with various amounts of training and experience with animals in general and abused animals in particular, and shelter adopters, who likely have even less training or experience with animals, are surveyed for their opinions. These non-experts may or may not represent what the people on the "front lines" of shelters may believe are the behavioral signs of animal abuse, but if the goal of this research is to point towards an ethogram, then we need to look not at the opinions of non-experts, but in the actual behaviors of actual animals known to have been abused. In short, it would be most appropriate to determine which behaviors can reliably be correlated to animal abuse and then determine if those behavior are being appropriately considered by the staff and adopters who interacted with sheltered animals. Starting with the views of non-experts of various backgrounds only serves to lead research off course in this area.

2. On a related note, I question the reliance on the behavioral signs of child abuse as the basis for this study. For example, person-specific fear in a pet cat or dog may be a sign of physical abuse; however, it may also be a sign that the pet was not adequately socialized during their short but critical socialization window (feline - 2-8 weeks, canine 3-14 weeks) to a particular demographic of humans, as is frequently the case with animals taken from feral colonies, animals raised by inexperienced (not not abusive) breeders, and so on. Thus, while this sign may be quite reliable in children, it may be less so in pets whose life experiences can vary greatly from children. Further, the original source on child abuse was a continuing education article for nursing professionals (33). In short, we should not anthropomorphize animals.

3. The sample sizes were very small for this study, and there is a strong likelihood of voluntary response bias given the topic and the non-random manner in which participants were recruited. The results cannot be deemed to represent shelter workers or shelter adopters in general.

4. Many of the survey items reference vague descriptions that should have been more precisely defined. For example, terms such as "aggression,' "wary," and "demanding behavior," to name just a few, are not clear and could have many interpretations.

5. Considering signs of physical abuse in a shelter environment is exceptionally challenging, as the environment itself is a confounding factor in an animal's behavior. While, ultimately, shelters may be places where many abuse cases might be discovered, it would be more useful to study known abuse cases in home or foster settings where other confounding stressors might be better accounted for.

**Do you want your identity to be public for this peer review?** For information about this choice, including consent withdrawal, please see our Privacy Policy

Reviewer #1: No

Reviewer #3: No

---

## [Author Response · Author response to Decision Letter 2]

8 Jan 2026

Response to each point raised:

Reviewer's feedback:

Reviewer #1:

Reviewer #1: Research on emotional abuse in animals is essential for detecting and prosecuting cases of animal cruelty. The authors have strengthened the manuscript and successfully addressed all of my comments.

Thank you again for your review and valuable time.

Reviewer #3:

Thank you for your feedback and we now have more clearly brought forward two things. Firstly, we write more on our rationale for studying the two populations from which our samples originated. Secondly, we bring forward more strongly how we deem this study a small first step towards other studies that are of higher scientific strength, such as studies objectively observing animal behaviours, based on an ethogram and done so by trained observers. We feel that our present small-scale study will help to point at the importance of such studies and initiate such studies, which might be deemed challenging to set up and execute. Below we provide more details on the changes to the manuscript.

1. The participants in the study, shelter workers with various amounts of training and experience with animals in general and abused animals in particular, and shelter adopters, who likely have even less training or experience with animals, are surveyed for their opinions. These non-experts may or may not represent what the people on the "front lines" of shelters may believe are the behavioral signs of animal abuse, but if the goal of this research is to point towards an ethogram, then we need to look not at the opinions of non-experts, but in the actual behaviors of actual animals known to have been abused. In short, it would be most appropriate to determine which behaviors can reliably be correlated to animal abuse and then determine if those behavior are being appropriately considered by the staff and adopters who interacted with sheltered animals. Starting with the views of non-experts of various backgrounds only serves to lead research off course in this area.

Thank you for highlighting a concern that this first step should not lead to inclusion or exclusion of possible behaviours in an ethogram for studying the likeliness of an animal’s possible historic abuse experience. We have rephrased sentences to ensure this is not unintentionally suggested and that we merely intend to point at the importance of such ethogram-based studies and the need to initiate such studies. E.g. in line 401 onwards: ‘Our study is a first step towards identifying behavioural signs of animal abuse. Our study findings cannot be used to assess animal abuse based on behavioural signs. We stress that the surveying of our particular study populations of shelter animal staff and adopters, comes with a limited use of our results. When constituting an ethogram of behaviours that may serve as possible abuse signs, our present study should not be used to determine the inclusion or exclusion of possible behaviours. We did not work with trained observers and did not study the animals themselves. Value of our study is in having gathered a first insight into our sample’s views on possible abuse signs. The findings indicate the relevance of a future possibility for such assessment. By establishing deemed relevance, we have not yet established deemed value of behavioural signs in animal abuse assessment. Behaviours of interest need to be identified and studied in observational studies of animals, preferably comparing the behaviour of animals that have suffered animal abuse to that of animals that have not suffered animal abuse. Once behavioural differences have objectively been established the behaviours will next need to be validated for their potential use to indicate animal abuse.’

We have more clearly indicated why we opted for shelter staff and shelter animal adopters as study populations, from line 94 onwards: ‘We have two reasons for involving shelter staff and animal adopters. The first reason is that abused animals form a part of the animal population under their care, as indicated in the previous paragraph [37]. This makes for an accessible population that may have under their care the animals of interest. Other such populations, are difficult to reach, due to it for example being unsafe or unethical to survey domestic abuse victims who have or had animals under their care, or to study animals seized, but not released and possibly involved in a legal trajectory. Also, deceased animals cannot be studied for behaviour, which is a contrast to studies assessing physical abuse signs. The second reason is that shelter staff and adopters may have views on possible previous abuse of animals under their care. These views are of interest as presently little is known on the validity of such views in regard to appointing an animal’s previous abuse or lack thereof. Learning more about these views can help us to not wrongfully appoint or reject a history of animal abuse.’

2. On a related note, I question the reliance on the behavioral signs of child abuse as the basis for this study. For example, person-specific fear in a pet cat or dog may be a sign of physical abuse; however, it may also be a sign that the pet was not adequately socialized during their short but critical socialization window (feline - 2-8 weeks, canine 3-14 weeks) to a particular demographic of humans, as is frequently the case with animals taken from feral colonies, animals raised by inexperienced (not not abusive) breeders, and so on. Thus, while this sign may be quite reliable in children, it may be less so in pets whose life experiences can vary greatly from children. Further, the original source on child abuse was a continuing education article for nursing professionals (33). In short, we should not anthropomorphize animals.

We feel there is relevance in studying behavioural signs of child abuse for their possible relevance in animals. Humans are known to show comparisons (though not similarity) in how they approach children and cats/dogs. E.g. attachment bonds and parenting styles apply to the interhuman relationships as well as to the relations between humans and cats/dogs to a degree1-6. We agree that we should not anthropomorphize animals. To prevent anthropomorphism, information is necessary on whether people, such as shelter animal staff and adopters, may or may not be inclined to do so. We do not feel that our study should be an end point. Contrary: we agree that other methods are highly necessary to assess which behavioural signs may or may not be relevant tot indicating an animal’s likely historic abuse, or lack thereof. We have strengthened the information on that point, as indicated under point 1) above. Additionally, we suggest future studies to consider training animal caretakers if these would be questioned on animal abuse behavioural signs (line 413 onwards): ‘(…). It is likely that in these studies seized sheltered animals will be studied. Their caretakers may be asked to report on animal behaviour, including on possible behavioural signs of animal abuse. Our study highlights that shelter animal staff and adopters may differ in their recognition of behaviours other than fear, when assessing likeliness of animal abuse. Consequently, either training caretakers involved in animal behaviour studies in their reporting of animal abuse, and/ or using additional validation sources, may benefit the quality of studies on animal abuse.’

1. Brubaker L, Udell MA. Does Pet Parenting Style predict the social and problem-solving behavior of pet dogs (Canis lupus familiaris)?. Animal cognition. 2023 Jan;26(1):345-56.

2. Kuo CH, Kessler S. Intergenerational transmission of human parenting styles to human–dog relationships. Animals. 2024 Mar 28;14(7):1038.

3. Luchesi SH, Machado DS, Trindade PH, Mikulincer M, Otta E. Psychometric validation of the Brazilian Version of the Pet Attachment Questionnaire (PAQ): An examination of predictors of attachment styles among cat owners. Applied Animal Behaviour Science. 2022 Nov 1;256:105769.

4. Rehn T, Beetz A, Keeling LJ. Links between an owner’s adult attachment style and the support-seeking behavior of their dog. Frontiers in psychology. 2017 Nov 30;8:2059.

5. Stoeckel LE, Palley LS, Gollub RL, Niemi SM, Evins AE. Patterns of brain activation when mothers view their own child and dog: An fMRI study. PloS one. 2014 Oct 3;9(10):e107205.

6. Volsche S, Gray P. “Dog moms” use authoritative parenting styles. Human-animal interaction bulletin. 2016 Dec 1(2016).

3. The sample sizes were very small for this study, and there is a strong likelihood of voluntary response bias given the topic and the non-random manner in which participants were recruited. The results cannot be deemed to represent shelter workers or shelter adopters in general.

We agree and have strengthened text sections addressing this limitation of our study, such as in the abstract in line 49-51: ‘Studying such differences in larger samples may provide insight into how these populations assess an animal’s possible abuse experience or lack thereof.’ and the discussion section, line 376 onwards: ‘Due to this limited number of participants, we caution against extrapolation of our findings to the total (Dutch) population of shelter staff and animal adopters.’. Also, we deemed in necessary from the start to stress the low numbers, such as in the abstract in line 39.

4. Many of the survey items reference vague descriptions that should have been more precisely defined. For example, terms such as "aggression,' "wary," and "demanding behavior," to name just a few, are not clear and could have many interpretations.

Thank you for allowing us to explain that we opted to stay close to the original wording used in the child abuse studies that formed the basis of this study. We do not see our study as having the scientific strength that an objective animal behaviour observation study would have. We argument at several points in our manuscript the necessity of this studying. However, we feel that our study increases the chance that such studies will be set up and executed. We have strengthened the section in the discussion on this particular limitation of our study, in line 378 onwards: ‘Also, we deemed it necessary to stay close to the approach taken in the child abuse studies that formed the basis of our study. This allows for the child abuse study outcomes to be compared to the outcomes of this study. Consequently, we opted not to work with objective descriptions of animal behavioural signs. Thus, there is a need for such objectively defined behavioural descriptions, specified in an ethogram, to form the basis of an objective assessment of animal abuse signs in future studies. To date, when animal abuse literature describes behavioural signs, these are often described in general terminology, without behavioural specifications, such as ‘signs of distress’, ‘the animal displays an unexplained change in behaviour’, or ‘observations of normal/ abnormal behaviour’ [6]. Establishing an ethogram that includes behaviours that objectively and discriminatively describe those behaviours that may be of interest as behavioural signs of animal abuse is one of the many steps to be taken in future studies.’.

5. Considering signs of physical abuse in a shelter environment is exceptionally challenging, as the environment itself is a confounding factor in an animal's behavior. While, ultimately, shelters may be places where many abuse cases might be discovered, it would be more useful to study known abuse cases in home or foster settings where other confounding stressors might be better accounted for.

Thank you for your interesting suggestion to set up a study with these particular study setting characteristics. We agree that shelter settings and shelter animal adoptions come with challenges and this underpinned our second reason to aim at gaining insight into e.g. shelter staff views. This may aid to take such factors into account in future, when assessing animal abuse. We discuss how people’s views (that is in our study the views of shelter staff and shelter animal adopters) may factor in as confounders, e.g. in line 355 onwards: ‘Additionally, expectations of why animals are shelter-surrendered may affect shelter staff and animal adopters differently. We know of no studies comparing such expectations between these two groups, but some studies into shelter animal adopters’ experiences are available. These studies indicate prior concerns of those contemplating to adopt a cat or dog, such as regarding an unknown history of the animal or their unwanted behaviours [47]. Such concerns may be valid, as 72% of N=192 adopted dogs reportedly showed undesired behaviour in the first week after adoption, such as aggression (24%) and fearfulness (21%) [37]. Undesired behaviour was reported at higher rates for dogs that suffered ‘abuse’ (no precise definition for abuse was provided in this study; [37]). After half a year the rates of reported behaviour were lower for fear, but not for aggression [37], and aggression was listed as a return reason for adopted cats and dogs [48]. Often provided reasons to opt for adoption when contemplating to acquire a cat or dog are for it to be ethically or morally correct [49]. So, when a choice for a shelter adoption is made ‘to do good’ and the animal shows undesired behaviours, expectations of a rescued animal ‘being grateful’ and ‘good’ may be unfulfilled. Consequently, it may be helpful to explain such unfulfilled expectations through a previous owner mistreating the animal and behavioural signs may more readily be viewed as a sign of animal abuse. (…)’.

---

## [Editor Report · Decision Letter 2]

19 Jan 2026

Dear Dr. Herwijnen,

Thank you for submitting your manuscript to PLOS ONE. After careful consideration, we feel that it has merit but does not fully meet PLOS ONE’s publication criteria as it currently stands. Therefore, we invite you to submit a revised version of the manuscript that addresses the points raised during the review process.

We look forward to receiving your revised manuscript.

Kind regards,

Brittany N. Florkiewicz, Ph.D.

Academic Editor

PLOS One

Journal Requirements:

Additional Editor Comments (if provided):

Thank you for addressing the comments from the previous reviewers. At this point, there are a few areas of the manuscript that could benefit from grammatical checks and rewording, particularly in the introduction. Additionally, the introduction requires clarification, which will assist in addressing the reviewer's comments regarding study limitations. Below, you will find my suggestions. I also recommend consulting a program or service to assist with these grammatical changes. Once these revisions are made, I will review the manuscript again and proceed to accept it for publication.

Line 57: Please rephrase, as the current wording is confusing. Perhaps something along the lines of..."Artificial selection has resulted in various traits that can sometimes negatively impact the overall well-being of domesticated species".

Lines 60-61: If you argue that animal abuse exists in our “societies”, please support your claim with data from multiple countries.

Lines 61-63: Are you suggesting that most people view animal abuse negatively and consider animals to be the victims in these situations?

Lines 63-64: It's difficult to determine whether your study is specifically focused on physical abuse, as you are only surveying shelter workers and prospective adopters about their general perceptions of whether the animals have been abused in the past. Additionally, you're asking which signs they believe to be the most reliable indicators of abuse. For this reason, you should consider addressing all aspects of animal abuse in your review.

Lines 67-68: I believe this can be combined with the previous sentence for clarity. “Various definitions of animal abuse exist; however, in our current study, we define it as...”

Line 78-79: This undermines your study because it focuses solely on the behaviors that shelter workers and staff associate with abuse. Additionally, based on the citations provided, there has been some research conducted on this topic. I recommend omitting the current content and summarizing what is known about behavioral indicators so far, even if the information is limited.

Lines 81-89: I would rephrase this paragraph to emphasize that it is easier to distinguish behavioral indicators of abuse in shelter settings compared to other contexts. Right now, the wording is unclear.

Line 87-89: To be cautious, I would recommend softening this language a bit, especially since there may be data reports available that are not published in scientific journals. This highlights a common issue in research on domesticated animal behavior, which could be worth mentioning.

Lines 91-104: Similar to my previous comment, please clarify that your initial focus is on assessing whether signs of animal abuse are perceived as significant by staff members and potential adopters. In the second step, you will evaluate which signs they consider to be the most prominent. Many studies compare the accuracy of veterinarians' perceptions with those of non-veterinarians regarding animal behavior. These studies should be cited to support the idea that experience results in higher accuracy.

Lines 92-93: You need to better explain the rationale behind linking child abuse to animal abuse in the current study. Additionally, the information from lines 74-79 could be integrated here, as it currently seems out of place.

Line 96: Remove “as indicated in the previous paragraph”. Please combine the sentences from lines 95-96 with the sentence from lines 96-97 for clarity.

Line 106: Just state “Our current study will focus on domesticated dogs and cats”.

Lines 106-112: Clearly specify how the behavioral indicators of fear manifest. Additionally, for "other" behaviors, aim to be more specific (e.g., prosocial behaviors, aggressive behaviors, etc.). Consult lines 309-313, as it appears you included this information in the discussion but not the introduction.

Lines 122-124: Please place this before the previous paragraph.

Lines 320-321: Remove this sentence.

Lines 340-342: Reword for clarity.

Lines 401-402: Please reword it as follows: “Our study represents an initial step in identifying behavioral signs of abuse that are noticeable to human observers. However, it is important to clarify that our research does not evaluate the accuracy of human assumptions in relation to the actual experiences of the animal”. You need to accurately portray your study, which focuses on human perceptions. Also, please add this caveat to your introduction.

---

## [Author Response · Author response to Decision Letter 3]

28 Jan 2026

Utrecht, January 24th, 2026

Dear Academic Editor and Reviewers,

Thank you for the new review of our manuscript titled ‘Abuse recognition by shelter staff and shelter animal adopters’. Please find below our response to the points that were raised.

Sincerely,

Ineke R. van Herwijnen Ph.D.

i.r.vanherwijnen@uu.nl

Division of Animals in Science and Society, Faculty of Veterinary Medicine, Department of Population Health Sciences, Utrecht University, Utrecht, The Netherlands

Response to each point raised:

Thank you for addressing the comments from the previous reviewers. At this point, there are a few areas of the manuscript that could benefit from grammatical checks and rewording, particularly in the introduction. Additionally, the introduction requires clarification, which will assist in addressing the reviewer's comments regarding study limitations. Below, you will find my suggestions. I also recommend consulting a program or service to assist with these grammatical changes. Once these revisions are made, I will review the manuscript again and proceed to accept it for publication.

Thank you and we have made grammatical changes, particularly in the introduction section.

Line 57: Please rephrase, as the current wording is confusing. Perhaps something along the lines of..."Artificial selection has resulted in various traits that can sometimes negatively impact the overall well-being of domesticated species".

The text was adapted.

Lines 60-61: If you argue that animal abuse exists in our “societies”, please support your claim with data from multiple countries.

Thank you for this suggestion, we added references from multiple countries.

Lines 61-63: Are you suggesting that most people view animal abuse negatively and consider animals to be the victims in these situations?

We have replaced the word ‘victims’.

Lines 63-64: It's difficult to determine whether your study is specifically focused on physical abuse, as you are only surveying shelter workers and prospective adopters about their general perceptions of whether the animals have been abused in the past. Additionally, you're asking which signs they believe to be the most reliable indicators of abuse. For this reason, you should consider addressing all aspects of animal abuse in your review.

We are specifically interested in perceptions of physical abuse and have stated this at several point in the manuscript, also explaining why we have this study interest. Previous review led to removal of information on alternative forms of animal abuse, such as emotional animal abuse, as to not confuse readers and avoid unintentionally indicating a broader scope of our study.

Lines 67-68: I believe this can be combined with the previous sentence for clarity. “Various definitions of animal abuse exist; however, in our current study, we define it as...”

We have made this adaptation.

Line 78-79: This undermines your study because it focuses solely on the behaviors that shelter workers and staff associate with abuse. Additionally, based on the citations provided, there has been some research conducted on this topic. I recommend omitting the current content and summarizing what is known about behavioral indicators so far, even if the information is limited.

We have made changes that should more clearly communicate what we aimed to state with this section.

Lines 81-89: I would rephrase this paragraph to emphasize that it is easier to distinguish behavioral indicators of abuse in shelter settings compared to other contexts. Right now, the wording is unclear.

We have adapted this text section. Unfortunately, we cannot yet say that it is easier to distinguish behavioural indicators of abuse in shelter settings compared to other contexts – such comparative studies are to our knowledge non-existent. We agree with one of the previous reviewers who pointed this out.

Line 87-89: To be cautious, I would recommend softening this language a bit, especially since there may be data reports available that are not published in scientific journals. This highlights a common issue in research on domesticated animal behavior, which could be worth mentioning.

We have reworded these lines.

Lines 91-104: Similar to my previous comment, please clarify that your initial focus is on assessing whether signs of animal abuse are perceived as significant by staff members and potential adopters. In the second step, you will evaluate which signs they consider to be the most prominent. Many studies compare the accuracy of veterinarians' perceptions with those of non-veterinarians regarding animal behavior. These studies should be cited to support the idea that experience results in higher accuracy.

We have added the information on the steps, while trying to maintain the text requested by a previous reviewer. Adding literature on veterinarian’s perceptions, would come with a risk of wrongfully suggesting that our present study, with a small number of participants only, can already be used for purposes such as assessing an effect of experience level in shelter staff.

Lines 92-93: You need to better explain the rationale behind linking child abuse to animal abuse in the current study. Additionally, the information from lines 74-79 could be integrated here, as it currently seems out of place.

We added this text as to adhere to a previous reviewer’s comments, but have made adaptations in both text sections to improve the flow in the introduction.

Line 96: Remove “as indicated in the previous paragraph”. Please combine the sentences from lines 95-96 with the sentence from lines 96-97 for clarity.

This text section has been removed.

Line 106: Just state “Our current study will focus on domesticated dogs and cats”.

This sentence was adapted.

Lines 106-112: Clearly specify how the behavioral indicators of fear manifest. Additionally, for "other" behaviors, aim to be more specific (e.g., prosocial behaviors, aggressive behaviors, etc.). Consult lines 309-313, as it appears you included this information in the discussion but not the introduction.

This text section was extended.

Lines 122-124: Please place this before the previous paragraph.

This section was replaced to the start of the Ethics section.

Lines 320-321: Remove this sentence.

We have removed the sentence: ‘As such, attachment/ attention-seeking was of interest to us.’

Lines 340-342: Reword for clarity.

Thank you, we have rewritten this section.

Lines 401-402: Please reword it as follows: “Our study represents an initial step in identifying behavioral signs of abuse that are noticeable to human observers. However, it is important to clarify that our research does not evaluate the accuracy of human assumptions in relation to the actual experiences of the animal”. You need to accurately portray your study, which focuses on human perceptions. Also, please add this caveat to your introduction.

We have replaced this sentence and added the information at the end of the Introduction section also.

---

## [Editor Report · Decision Letter 3]

1 Feb 2026

Abuse recognition by shelter staff and shelter animal adopters

PONE-D-25-49955R3

Dear Dr. Herwijnen,

We’re pleased to inform you that your manuscript has been judged scientifically suitable for publication and will be formally accepted for publication once it meets all outstanding technical requirements.

Kind regards,

Brittany N. Florkiewicz, Ph.D.

Academic Editor

PLOS One
---

## [Editor Report · Acceptance letter]

PONE-D-25-49955R3

PLOS One

Dear Dr. Herwijnen,

I'm pleased to inform you that your manuscript has been deemed suitable for publication in PLOS One. Congratulations! Your manuscript is now being handed over to our production team.

Kind regards,

on behalf of

Dr. Brittany N. Florkiewicz

Academic Editor

PLOS One